# Training Matryoshka Mixture-of-Experts for Elastic Inference-Time Expert Utilization

## Abstract

Mixture-of-Experts (MoE) has emerged as a promising paradigm for efficiently scaling large language models without a proportional increase in computational cost. However, the standard training strategy of Top-K router prevents MoE models from realizing their full potential for elastic inference. When the number of activated experts is altered at inference time, these models exhibit precipitous performance degradation. In this work, we introduce Matryoshka MoE (M-MoE), a training framework that instills a coarse-to-fine structure directly into the expert ensemble. By systematically varying the number of activated experts during training, M-MoE compels the model to learn a meaningful ranking: top-ranked experts collaborate to provide essential, coarse-grained capabilities, while subsequent experts add progressively finer-grained detail. We explore this principle at multiple granularities, identifying a layer-wise randomization strategy as the most effective. Our experiments demonstrate that a single M-MoE model achieves remarkable elasticity, with its performance at various expert counts closely matching that of an entire suite of specialist models, but at only a fraction of the total training cost. This flexibility not only unlocks elastic inference but also enables optimizing performance by allocating different computational budgets to different model layers. Our work paves the way for more practical and adaptable deployments of large-scale MoE models.

## 1 Introduction

The landscape of artificial intelligence is increasingly dominated by large-scale models (OpenAI, 2023; DeepSeek-AI, 2025; google, 2025), whose unprecedented capabilities (Scale-AI, 2025) are often shadowed by their immense computational cost. This has given rise to a critical need for elastic inference (Cai et al., 2024): the ability of a single model to dynamically adapt its computational footprint to meet diverse user requirements.

A prominent and successful paradigm in this domain is Matryoshka Representation Learning (MRL)(Kusupati et al., 2022) and its architectural derivatives(Devvrit et al., 2024; GemmaTeam, 2025). MRL addresses a fundamental inefficiency in deep learning: standard models tend to diffuse information evenly across their entire representation vectors, which makes smaller, truncated representations ineffective. MRL directly counteracts this by instilling a structured, coarse-to-fine granularity within a single high-dimensional embedding. The training objective is applied not only to the full representation but also to its nested, truncated prefixes. This forces the model to prioritize and pack the most critical, high-level information into the initial dimensions, with subsequent dimensions progressively adding finer-grained detail.

While MRL explicitly instills a Matryoshka structure within a single representation, Mixture-of-Experts (MoE) architectures (Shazeer et al., 2017) present an innate structural potential for the same principle. As the leading paradigm for scaling models to billions of parameters at a manageable computational cost (Jiang et al., 2024), MoE routes each token through a small subset of expert subnetworks. Instead of adapting model depth or representation dimension, MoE's sparse architecture naturally suggests adapting its width—the number of concurrently active experts. The intuition is powerful: at inference time, one could simply select fewer experts for a coarse but fast prediction, or more experts for a fine-grained, higher-quality output, effectively creating a "Matryoshka MoE".

However, our empirical investigation into publicly available MoE models reveals a counter-intuitive reality. As shown in figure 1, increasing the number of activated experts yields minimal performance gains, while reducing it leads to progressively accelerating performance degradation. This finding directly contradicts the prevailing intuition and exposes a fundamental brittleness in current MoE models, suggesting they are incapable of delivering on the promise of true inference-time elasticity. This performance collapse stems from the inherent rigidity of the fixed Top-K training paradigm. During training, each expert becomes overly specialized in collaborating with a fixed-size group of peers. this paradigm causes a problem analogous to information diffusion: expert capacity becomes rigidly co-adapted to a fixed-size group, and the router's ranking ability is only meaningful for the top K. Any deviation disrupts this delicate balance.

To overcome this critical limitation and unlock the true potential of MoE for elastic inference, we propose Matryoshka MoE (M-MoE), a simple yet effective training strategy. The core idea of M-MoE is to instill a coarse-to-fine granularity within the MoE's expert routing mechanism. We explore this principle at different granularities, ranging from randomizing the expert count for an entire global batch to our most effective strategy: a layer-wise approach where each Transformer layer independently selects a different number of experts. This fine-grained stochasticity forces experts to differentiate their contributions, with fewer activated experts collaboratively providing essential, coarse-grained information, and additional experts progressively adding finer-grained detail, thereby fostering a more versatile model.

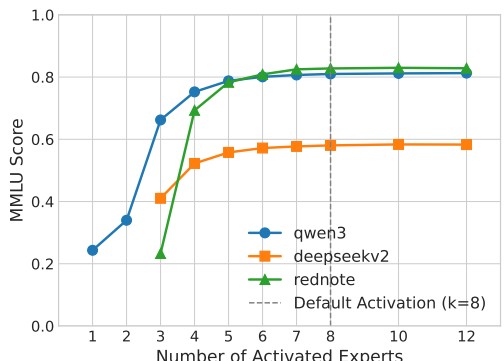

Figure 1: MMLU score of DeepSeek-V2-Lite, Qwen3-30B-A3B-Base, and RedNote-Dots.LLM1.Base under varying numbers of activated experts.

Our experiments demonstrate that a single M-MoE model can achieve remarkable inference-time elasticity, delivering performance that is comparable to multiple specialist models, each trained individually for a specific expert count. The analysis of the router's internal mechanics reveals that M-MoE not only teaches the gating network to produce a globally coherent and stable ranking of experts, but also fosters a higher degree of expert specialization. This is in stark contrast to the brittle rankings and greater functional overlap among experts observed in fixed-k models. Furthermore, the inherent flexibility of our M-MoE model unlocks novel analytical possibilities. We investigate the performance impact of allocating different numbers of experts to different layers during inference, providing valuable insights for future elastic deployment strategies, which is impossible with rigidly trained MoE models.

Our main contributions are as follows:

- We identify the rigidity of fixed-k training as a key barrier to elastic MoE inference and propose Matryoshka MoE, a framework that instills a coarse-to-fine functional hierarchy within the expert ensemble.

- We empirically demonstrate that our layer-wise M-MoE strategy is highly effective, producing a single, elastic model that rivals the performance of an entire suite of specialist models at a fraction of the training cost.

- Through detailed analysis, we show that M-MoE induces stable expert rankings and functional specialization, unlocking the ability to analyze and deploy novel layer-wise inference strategies.

## 2 PRELIMINARY

In this section, we provide an overview of the Mixture-of-Experts (MoE) architecture and its standard routing mechanism, which form the foundation of our work.

## 2.1 MIXTURE-OF-EXPERT TRANSFORMERS

The Mixture-of-Experts (MoE) paradigm is a powerful architectural innovation for efficiently scaling Large Language Models. In the context of the Transformer architecture, which forms the backbone of modern LLMs, MoE is typically implemented by replacing the standard, dense feed-forward network (FFN) sublayer within each Transformer block. This modification transforms the FFN into a collection of $N$ independent expert networks $\{E_1, E_2, \ldots, E_N\}$, each retaining the original FFN's structure.

Accompanying the set of experts is a lightweight gating network, or router, $G$. For each input token $\mathbf{x}$, the router dynamically selects a sparse subset of these experts to process the token. The final output of the MoE layer, $\mathbf{y}$, is a weighted combination of the outputs from the selected experts:

$$\mathbf{y} = \sum_{i=1}^{N} w_i \cdot E_i(\mathbf{x}), \tag{1}$$

where $w_i$ is the weight assigned by the router to the $i$-th expert. In sparsely-gated MoEs, most of these weights are zero, ensuring that only a small fraction of experts are computationally active for any given input. This conditional computation allows MoE models to possess a vast number of parameters without a proportional increase in FLOPs, enabling significant gains in model capacity and performance (Rajbhandari et al., 2022).

## 2.2 TOP-K ROUTING

The most prevalent mechanism for implementing the sparse selection in MoE models is Top-k routing (Shazeer et al., 2017; Jiang et al., 2024; Yang et al., 2025). Given an input token $\mathbf{x}$, the gating network $G$ first computes a logit score $s_i$ for each of the $N$ experts, typically via a linear transformation followed by a softmax function:

$$\mathbf{s} = \text{Softmax}(\mathbf{x} \cdot \mathbf{W}_g), \tag{2}$$

where $\mathbf{W}_g$ is the learnable weight matrix of the router. The softmax can also be replaced with a sigmoid function.

The Top-k routing strategy then selects the $k$ experts corresponding to the highest scores in $\mathbf{s}$. Let $\mathcal{T}$ be the set of indices of these top $k$ experts. The weights $w_i$ from Equation 1 are then defined as:

$$w_i = \begin{cases} \frac{s_i}{\sum_{j \in \mathcal{T}} s_j} & \text{if } i \in \mathcal{T} \\ 0 & \text{if } i \notin \mathcal{T} \end{cases} \tag{3}$$

The value of $k$ is a critical hyperparameter that remains fixed throughout both the training and inference phases.

## 3 MATRYOSHKA MOE

The core principle of Matryoshka MoE (M-MoE) is to introduce diversity into the number of activated experts during training, therefore compelling the router to learn a truly meaningful ranking: the top-ranked experts are incentivized to capture the most essential, high-level information, while progressively lower-ranked experts contribute increasingly fine-grained specializations. In this section, we explore several distinct strategies for implementing this principle.

### 3.1 BATCH-LEVEL MATRYOSHKA

The foundational Matryoshka strategy is to randomize the expert count, $k$, at the batch level. This can be implemented at two distinct granularities, reflecting different trade-offs between randomization frequency and implementation simplicity in distributed training.

The first granularity is the **global batch**, which represents the entire dataset processed for a single optimizer step. In this setting, one value of $k$ is sampled and applied uniformly to all data within that global batch. The second, more dynamic granularity is the **micro-batch**. A micro-batch corresponds

to the subset of data consumed in a single forward pass by a model instance. Here, a new value of $k$ can be sampled for each micro-batch, introducing a higher frequency of variation.

For either granularity, a single value $k_{\text{dyn}}$ is drawn from a uniform distribution:

$$k_{\text{dyn}} \sim \mathcal{U}[k_{\min}, k_{\max}]. \tag{4}$$

This $k_{\text{dyn}}$ is then applied to all tokens and all MoE layers within the designated batch (global or micro). While Batch-level M-MoE establishes a crucial baseline, it enforces a uniform expert count across all layers, a rigid constraint that we address next.

## 3.2 LAYER-WISE MATRYOSHKA

To better accommodate the functional specialization of different model layers and prevent over-specialization, we propose **Layer-wise M-MoE**. This advanced strategy decouples the choice of the number of active experts, $k$, across different layers, pushing the Matryoshka principle to its full potential. For any given token, each MoE layer is free to activate a different number of experts, forcing the representations at each stage of the network to be robust to varying computational widths from the preceding layer. This maximal stochasticity hypothesizes that different layers may benefit from different levels of expert capacity. We explore two primary strategies for sampling the per-layer expert count, $k_l$.

**Uniform Sampling.** The most straightforward implementation of layer-wise stochasticity is to sample $k_l$ for each layer $l$ from a discrete uniform distribution:

$$k_l \sim \mathcal{U}[k_{\min}, k_{\max}]. \tag{5}$$

This approach treats all possible expert counts as equally likely, ensuring a broad and unbiased exploration of the elasticity space during training. It serves as our baseline strategy, designed to build a general-purpose elastic model without making prior assumptions about which expert configurations are more important to learn.

**Capacity-Aware Weighted Sampling.** The principle of uniform sampling might overlook a critical aspect of model scaling: configurations that activate more experts possess greater capacity and may require more extensive training to fully realize their potential (Tian et al., 2025). To account for this, we introduce a principled, temperature-controlled framework for Capacity-Aware Weighted Sampling.

We define a score for each expert count $k$, which is a simple monotonic function $f(k)$ reflecting its capacity. We then transform these scores into a probability distribution using a softmax function with a temperature parameter, $\tau$:

$$P(k_l = k) \propto \exp\left(\frac{f(k)}{\tau}\right). \tag{6}$$

By choosing $f(k) = \log(k)$, our sampling formula simplifies to a power law, $P(k_l = k) \propto k^{1/\tau}$, which provides an intuitive control over the distribution's shape. This principled framework allows for a systematic exploration of the trade-offs between providing sufficient training signal to high-capacity modes and ensuring robust performance across the entire elasticity spectrum.

## 3.3 ALTERNATIVE: PROBABILITY-BASED MATRYOSHKA

As an alternative approach that also embodies the Matryoshka principle of variable activation, we investigate probability-based routing. Following the work of Dynamic-MoE (Huang et al., 2024), we employ Top-p routing with a probability threshold, $p$. The number of activated experts, $k_p(\mathbf{x})$, is determined on a per-token basis by the router's confidence:

$$k_p(\mathbf{x}) = \min\left\{k' \mid \sum_{i=1}^{k'} \text{softmax}(\mathbf{s})_{(i)} \geq p\right\}, \tag{7}$$

where $\text{softmax}(\mathbf{s})_{(i)}$ are the router's softmax probabilities sorted in descending order. This method inherently introduces activation diversity and serves as an interesting point of comparison to our primary K-randomization based methods.

## 4 EXPERIMENT

We conduct a series of experiments to validate the effectiveness of the M-MoE framework. We compare our proposed methods against a standard Top-k baseline, first in a continual pre-training scenario and then from scratch. Besides, we explore the new possibilities unlocked by our layer-wise training strategy through fine-grained, layer-specific inference patterns. Finally, we present an in-depth analysis of the router's mechanics to show that M-MoE fosters superior expert specialization and ranking stability.

### 4.1 SETUP

**Model.** Our experiments are based on a 20-billion parameter MoE model. The architecture consists of 56 Transformer layers, each employing group query attention. The MoE layers, which replace the standard FFNs, contain a total of 96 experts. During a standard forward pass, only k experts are activated each layer, resulting in 0.5 billion active parameters when k = 1.

**Baselines and Methods.** We evaluate the following training strategies:

- **Top-k**: The standard baseline where the model is trained with a fixed number of activated experts.
- **Top-p**: As described in Section 3.3, the number of activated experts is calculated with a probability threshold.
- **M-MoE-global-batch**: As described in Section 3.1, where a single $k \in [1, 6]$ is sampled for each global training batch.
- **M-MoE-micro-batch**: $k \in [1, 6]$ is sampled for each micro-batch.
- **M-MoE-layer**: As described in Section 3.2, where each layer independently samples a $k \in [1, 6]$. All M-MoE strategies sample $k$ from a uniform distribution over the designated range, unless $\tau$ is specified.

**Training.** Our main experiments are conducted in a continual pre-training setting. We start from a base model that was pre-trained for 1T tokens, all layers activating a single expert. Starting from this checkpoint, we apply the different M-MoE strategies and the Top-k baseline for an additional training phase of 80B tokens. This setup simulates a practical scenario where one wishes to equip an existing MoE model with elastic capabilities without the prohibitive cost of retraining from scratch. Different from the setup in (Devvrit et al., 2024), where their elastic model was trained for a token count equivalent to the sum of its specialist baselines (e.g., 4x for 4 widths), we train our single M-MoE model for the same number of tokens as a single baseline. This provides a more challenging evaluation, as we simultaneously optimize for six different widths within the training budget of one.

More details about our model, training, data and evaluation can be found in Appendix B

### 4.2 MAIN RESULTS

The comprehensive results of our continual training experiments are presented in Table 1. The first section of the table shows the performance of specialist Top-k models. As expected, each model performs best at its native activation count (e.g., Top-k (k=6) at $k = 6$) but suffers a severe performance collapse when evaluated with a different number of experts, particularly when reducing to $k = 1$. This confirms the inherent rigidity of the standard training paradigm. In stark contrast, the M-MoE models, shown in the second section, demonstrate remarkable elasticity. They maintain strong performance across the entire spectrum of expert counts from $k = 1$ to $k = 6$. Critically, at lower activation counts ($k = 1$ and $k = 2$), all M-MoE variants significantly outperform the degraded specialist models.

Among the M-MoE strategies, **M-MoE-layer** consistently delivers the most robust performance, achieving the highest scores among the elastic models at nearly every evaluation point. This indicates that introducing stochasticity at the layer level is the most effective approach for learning versatile and generalizable expert representations, successfully realizing the goal of a single, truly elastic MoE model.

Table 1: Comprehensive performance comparison of specialist Top-k models and elastic M-MoE models. For each specialist model, performance at its native expert count is marked with an asterisk (*). Task abbreviations are: ARC-C (ARC Challenge), OBQA (OpenBookQA), WinoG. (Winogrande). For stable analysis, we bold and underline the best and second-best scores for each inference k value in the MMLU and Avg columns.

| Training Method | Inf. k | MMLU | ARC-C | BoolQ | HellaS. | LogiQA | OBQA | WinoG. | Avg |
|---|---|---|---|---|---|---|---|---|---|
| *Specialist Baselines (Top-k)* | | | | | | | | | |
| Top-k (k=1) | 1* | **52.01** | 52.22 | 67.43 | 70.38 | 29.95 | 40.00 | 67.64 | 54.23 |
| Top-k (k=2) | 1 | 35.54 | 32.59 | 65.08 | 49.09 | 30.11 | 32.60 | 61.17 | 43.74 |
| | 2* | 52.16 | 52.30 | 70.73 | 71.47 | 31.49 | 42.00 | 69.93 | 55.73 |
| Top-k (k=4) | 1 | 41.50 | 40.02 | 63.46 | 59.28 | 28.42 | 33.60 | 62.19 | 46.92 |
| | 2 | 50.42 | 51.62 | 66.33 | 70.41 | 29.03 | 41.20 | 68.67 | 53.95 |
| | 4* | 53.43 | 54.44 | 69.11 | 72.41 | 29.95 | 43.40 | 69.93 | 56.10 |
| Top-k (k=6) | 1 | 35.52 | 43.17 | 61.93 | 58.47 | 29.34 | 29.00 | 60.38 | 45.40 |
| | 2 | 48.90 | 52.17 | 62.75 | 69.87 | 29.11 | 38.40 | 66.38 | 52.51 |
| | 4 | 53.25 | 55.03 | 69.30 | 72.22 | 30.57 | 43.80 | 69.61 | 56.25 |
| | 6* | **54.32** | 55.46 | 70.92 | 72.88 | 31.34 | 43.80 | 70.48 | **57.03** |
| *Elastic Models (trained on k ∈ [1,6])* | | | | | | | | | |
| Top-p (p=0.1) | 1 | 35.54 | 43.01 | 63.09 | 55.52 | 27.96 | 30.40 | 60.06 | 45.08 |
| | 2 | 48.03 | 53.16 | 64.59 | 71.08 | 30.88 | 41.60 | 65.67 | 53.57 |
| | 4 | 52.46 | 54.86 | 68.50 | 72.15 | 30.57 | 42.60 | 69.06 | 55.74 |
| | 6 | 53.06 | 54.86 | 69.57 | 71.89 | 30.72 | 41.20 | 70.48 | 55.97 |
| M-MoE-global-batch | 1 | 50.78 | 51.54 | 67.52 | 71.01 | 29.34 | 41.20 | 67.96 | 54.19 |
| | 2 | 50.94 | 52.47 | 69.33 | 71.83 | 30.57 | 43.00 | 67.80 | 55.13 |
| | 4 | 52.27 | 54.01 | 70.70 | 72.28 | 30.11 | 43.40 | 69.53 | 56.04 |
| | 6 | 52.28 | 54.35 | 70.09 | 72.25 | 29.80 | 42.80 | 70.32 | 55.98 |
| M-MoE-micro-batch | 1 | 51.00 | 53.24 | 69.69 | 70.50 | 30.11 | 42.20 | 68.27 | **55.00** |
| | 2 | 51.64 | 53.75 | 69.69 | 71.74 | 28.42 | 42.60 | 68.03 | 55.12 |
| | 4 | 52.72 | 55.03 | 70.06 | 72.23 | 29.34 | 45.60 | 70.01 | 56.43 |
| | 6 | 52.89 | 54.78 | 69.91 | 72.05 | 29.80 | 43.20 | 70.40 | 56.15 |
| M-MoE-layer | 1 | 51.69 | 51.19 | 69.39 | 69.96 | 32.26 | 41.00 | 66.77 | 54.61 |
| | 2 | 52.71 | 53.50 | 71.44 | 72.43 | 31.49 | 42.40 | 68.82 | **56.11** |
| | 4 | 53.77 | 54.95 | 72.84 | 72.39 | 31.64 | 42.00 | 69.22 | 56.69 |
| | 6 | 53.56 | 54.95 | 72.72 | 71.70 | 31.95 | 43.00 | 69.14 | 56.72 |
| M-MoE-layer (τ=2) | 1 | 50.62 | 52.82 | 67.09 | 68.83 | 30.41 | 38.60 | 68.43 | 53.83 |
| | 2 | **53.42** | 53.75 | 67.43 | 72.41 | 31.80 | 43.00 | 70.40 | 56.03 |
| | 4 | **54.33** | 55.20 | 68.84 | 72.17 | 32.10 | 43.80 | 71.74 | **56.88** |
| | 6 | 54.14 | 55.55 | 69.36 | 71.79 | 32.10 | 43.80 | 71.82 | 56.94 |

## 4.3 FROM-SCRATCH PRE-TRAINING

While our primary focus is on the more practical continual training scenario, we also conducted an experiment to validate our approach when training from scratch. For this, we trained specialist Top-k models and a M-MoE-layer model for 80 billion tokens. The loss curve can be found in Appendix E.

The results, presented in Table 2, corroborate our main findings even at this earlier stage of training. The specialist models exhibit significant performance degradation when evaluated with only one active expert ($k = 1$), again demonstrating the brittleness of the fixed-k training paradigm. In contrast, the M-MoE-layer model shows remarkable robustness. This confirms that the benefits of the M-MoE training strategy are fundamental to the learning process and not merely an artifact of the continual training setup.

Regarding the observation that performance appears relatively flat across varying $k$ at the 80B checkpoint, we attribute this primarily to the early stage of pre-training. At this phase, training dynamics naturally favor the greedy optimization of top-ranked experts, limiting the immediate marginal contribution of lower-ranked experts. This trend is also visible in the specialist baselines. To investigate the long-term behavior of expert utilization, we extended the training of the M-MoE model to 160 billion tokens. As shown in the bottom section of Table 2, the 160B results demonstrate a clear upward trend in average performance as $k$ increases. This indicates that as the optimization efficiency

of dominant experts diminishes over time, the model successfully learns to leverage lower-ranked experts for further improvements, confirming that the additional experts are effectively utilized as the model matures.

## 4.4 LAYER-WISE INFERENCE

The M-MoE-layer model, which is trained with layer-decoupled stochasticity, unlocks the ability to deploy novel inference strategies where different parts of the model operate with different computational budgets. We explore this capability using our continually trained M-MoE-layer model. The model's 56 layers are divided into four sequential groups of 14 layers each.

Our investigation is centered on a baseline uniform activation pattern, [2, 2, 2, 2], which corresponds to activating 2 experts for every layer in the network. From this baseline, we explore how performance changes when we vary the number of active experts in specific layer groups, effectively redistributing the computational budget. As detailed in Table 3, we test two scenarios:

- **Increasing Capacity:** We evaluate patterns with a higher average of 2.5 experts per layer, distributing the additional capacity differently across the four layer groups.
- **Decreasing Capacity:** We test patterns with a lower average of 1.5 experts per layer to see which parts of the network are more resilient to a reduction in computation.

The results provide compelling evidence that earlier layers are more critical to the model's performance. When increasing the average expert count to 2.5, the [3, 3, 2, 2] configuration—which allocates extra experts to the first half of the model—achieves the most significant performance boost. Allocating the same extra capacity to the latter half ([2, 2, 3, 3]) results in a much smaller improvement over the baseline. This conclusion is reinforced when decreasing capacity. Reducing experts in the first half ([1, 1, 2, 2]) leads to a sharp performance drop. In contrast, configurations that preserve capacity in the early layers while reducing it in later ones) are far more robust. This asymmetry strongly suggests that for a given computational budget, prioritizing the capacity of earlier layers is a more effective optimization strategy.

Table 2: From-scratch pre-training results. We compare Specialist Baselines and M-MoE at 80B tokens, and provide extended training results for M-MoE at 160B tokens to demonstrate expert utilization. Best scores for each inference setting within 80B tokens are in **bold**.

| Training Method | Inf. k | MMLU | ARC-C | BoolQ | HellaS. | LogiQA | OBQA | WinoG. | Avg |
|---|---|---|---|---|---|---|---|---|---|
| | | | | 80B Tokens | | | | | |
| *Specialist Baselines (Top-k)* | | | | | | | | | |
| Top-k (k=1) | 1* | 27.91 | 35.75 | 57.13 | 52.41 | 27.04 | 35.20 | 51.30 | 40.96 |
| Top-k (k=2) | 1 | 24.74 | 26.19 | 39.14 | 26.42 | 24.12 | 25.40 | 50.43 | 30.92 |
| | 2* | **30.40** | 35.92 | 57.34 | 55.21 | 28.11 | 35.80 | 54.70 | **42.55** |
| Top-k (k=4) | 1 | 23.50 | 33.45 | 48.90 | 34.21 | 25.35 | 29.00 | 52.17 | 35.23 |
| | 2 | 28.86 | 35.86 | 47.52 | 54.28 | 28.26 | 35.20 | 53.99 | 40.57 |
| | 4* | **30.96** | 38.31 | 54.98 | 56.53 | 28.42 | 35.00 | 54.14 | **42.62** |
| Top-k (k=6) | 1 | 25.94 | 25.77 | 48.29 | 42.89 | 27.19 | 28.80 | 53.59 | 36.07 |
| | 2 | 25.49 | 36.60 | 44.80 | 54.88 | 28.11 | 35.00 | 53.83 | 39.82 |
| | 4 | 29.00 | 38.74 | 49.39 | 56.90 | 28.26 | 35.20 | 55.17 | 41.81 |
| | 6* | 30.20 | 38.99 | 54.39 | 56.97 | 29.80 | 35.20 | 56.59 | **43.16** |
| *Elastic Model (M-MoE)* | | | | | | | | | |
| **M-MoE-layer** | 1 | **28.71** | 36.52 | 56.27 | 52.75 | 26.27 | 33.60 | 53.43 | **41.08** |
| | 2 | 30.52 | 37.71 | 56.13 | 56.02 | 27.04 | 36.80 | 54.22 | 42.63 |
| | 4 | 30.76 | 38.91 | 55.25 | 56.08 | 27.80 | 35.80 | 54.06 | 42.67 |
| | 6 | **30.34** | 37.80 | 57.52 | 55.47 | 27.19 | 35.60 | 54.30 | 42.60 |
| | | | 160B Tokens (Extended Training) | | | | | | |
| **M-MoE-layer** | 1 | 32.94 | 43.34 | 62.51 | 59.60 | 28.88 | 34.40 | 56.83 | 45.50 |
| | 2 | 34.89 | 44.20 | 57.16 | 63.01 | 29.49 | 37.60 | 57.46 | 46.26 |
| | 4 | 35.24 | 44.37 | 60.31 | 63.12 | 28.73 | 37.60 | 58.56 | 46.85 |
| | 6 | 34.93 | 44.03 | 62.26 | 62.56 | 28.42 | 37.40 | 59.19 | 46.97 |

Table 3: Performance of the M-MoE-layer model under various layer-wise inference strategies. **Bold** scores indicate the largest performance deviation from the baseline.

| Activation Pattern | Avg. k | MMLU | ARC-C | BoolQ | HellaS. | LogiQA | OBQA | WinoG. | Avg |
|---|---|---|---|---|---|---|---|---|---|
| *Baseline* | | | | | | | | | |
| [2, 2, 2, 2] | 2.0 | 52.71 | 53.50 | 71.44 | 72.43 | 31.49 | 42.40 | 68.82 | 56.11 |
| *Increasing Capacity* | | | | | | | | | |
| [3, 3, 2, 2] | 2.5 | **53.35** | 53.33 | 71.01 | 72.92 | 32.26 | 43.00 | 68.98 | **56.41** |
| [2, 3, 3, 2] | 2.5 | 53.13 | 53.84 | 71.44 | 72.46 | 30.88 | 42.40 | 67.80 | 55.99 |
| [2, 2, 3, 3] | 2.5 | 52.90 | 53.75 | 71.74 | 72.25 | 30.88 | 41.80 | 68.98 | 56.04 |
| *Decreasing Capacity* | | | | | | | | | |
| [1, 1, 2, 2] | 1.5 | **51.63** | 53.75 | 69.88 | 71.51 | 31.20 | 41.40 | 68.59 | **55.42** |
| [2, 1, 1, 2] | 1.5 | 52.56 | 52.05 | 71.62 | 71.26 | 31.95 | 41.20 | 68.19 | 55.55 |
| [2, 2, 1, 1] | 1.5 | 52.78 | 52.90 | 71.07 | 71.23 | 31.95 | 40.60 | 68.11 | 55.52 |

## 4.5 MATRYOSHKA ROUTING

A core tenet of our Matryoshka MoE framework is that it should instill a nested, hierarchical structure within the expert routing mechanism. An ideal M-MoE router would learn a globally meaningful ranking where the Top-1 expert is the single most important contributor, the Top-2 experts form the best pair, and so on. This implies a critical property: the set of experts selected for a smaller budget ($k_{small}$) should be a proper subset of the experts selected for a larger budget ($k_{large}$). In contrast, a standard Top-k router is only trained to identify a good fixed-size team, and its ranking may become arbitrary outside that specific context.

To verify the existence of this nested routing behavior, we measure the consistency of the expert ranking across different budgets. We compute the Spearman rank correlation of router logits for a relevant set of experts—defined as the union of experts selected under a high budget ($k_{large} = 6$) and a low budget ($k_{small}$). We term this metric **Focused Spearman Correlation**.

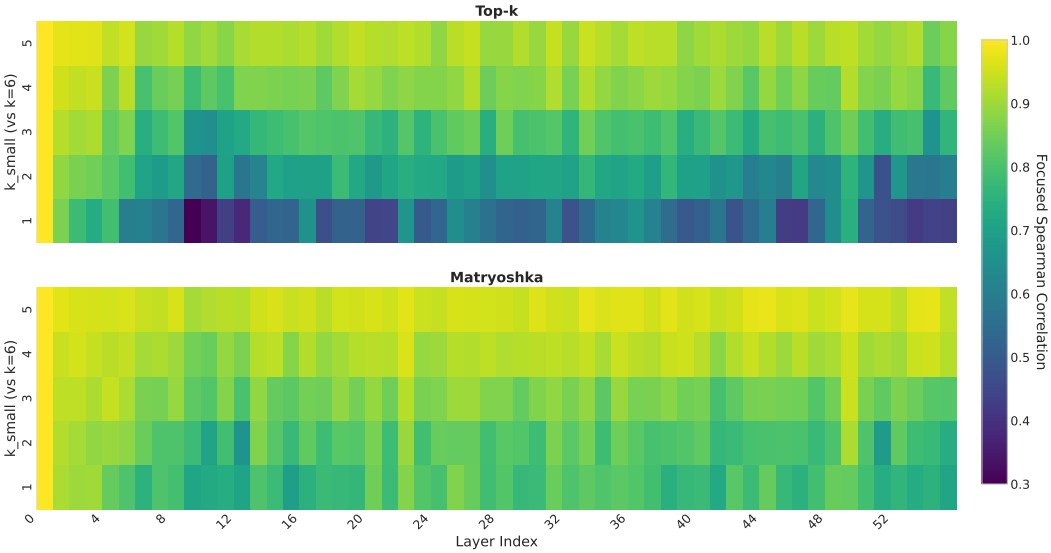

Figure 2: Heatmaps illustrating the router's expert ranking consistency for the Top-k ($k = 6$) model (top) and our M-MoE-Layer model (bottom). A bright color signifies a high correlation, indicating a strong nested, Matryoshka-like ranking structure.

We apply this analysis to both the baseline Top-k ($k = 6$) model and our M-MoE-Layer model, using $k_{large} = 6$ as the reference and varying $k_{small}$ from 1 to 5. The results, visualized in Figure 2, provide a stark contrast. The Top-k model (top panel) fails to exhibit a Matryoshka structure. The correlation plummets as $k_{small}$ deviates from its trained value of 6, turning the heatmap dark. Con-

versely, the M-MoE-Layer model (bottom panel) demonstrates a remarkably strong and consistent Matryoshka property. The heatmap remains bright across nearly all layers and values of $k_{small}$. This is compelling evidence that M-MoE training forces the router to learn a coherent, global, and hierarchical ordering of its experts. This learned nested structure is the fundamental mechanism behind the model's elasticity: activating more experts is analogous to revealing the next layer of a Matryoshka doll, with each additional expert building upon the coarse-grained foundation provided by the smaller, nested set. This property underpins the model's ability to gracefully scale its performance with its computational budget.

### 4.6 Expert Specialization

We hypothesize that the Top-k paradigm may inadvertently encourage functional overlap among experts, whereas the variability of M-MoE training should foster greater specialization. To investigate this, we analyze the geometric relationships between the router's gating weights. Each expert's gating weight can be viewed as a vector in a high-dimensional space, whose direction signifies the expert's preferred input features. A high degree of specialization implies that different experts should attend to different features, meaning their corresponding weight vectors should be as orthogonal as possible. We quantify this specialization using the Mean Off-Diagonal Similarity (MODS). For each MoE layer, we compute the cosine similarity matrix of its L2-normalized expert gating vectors. The MODS is then defined as the average of the absolute values of the off-diagonal elements of this matrix. A low MODS value signifies high orthogonality and thus a high degree of expert specialization.

We applied this analysis to both the Top-k (k=6) baseline and our M-MoE-Layer model. The results, plotted across all MoE layers, are presented in Figure 3. The results reveal a significant and consistent advantage for our M-MoE-Layer model. As shown in the figure, its MODS curve is markedly lower than that of the Top-k baseline across nearly the entire depth of the network. This provides strong quantitative evidence that M-MoE training successfully cultivates a set of more distinct and specialized experts. The reduced similarity implies that each expert has carved out a more unique functional niche, minimizing redundancy within the model.

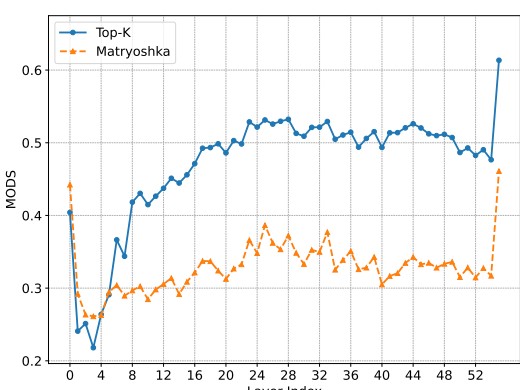

Figure 3: Comparison of MODS for the Top-k and our model. Lower MODS indicates greater expert specialization.

Interestingly, both models exhibit a similar overall trend: the MODS value is relatively high at the first MoE layer, experiences a sharp drop in the subsequent layers, and then gradually increases towards the end of the model. This may suggest that experts in the initial layers handle more general, foundational tasks, while specialization peaks in the middle layers. Towards the final layers, a degree of functional convergence might be necessary to integrate complex features for the final output.

## 5 Related Work

### 5.1 Expert Routing

While Top-K routing is the standard practice in numerous state-of-the-art LLMs (DeepSeek-AI, 2025; Yang et al., 2025), its inherent rigidity has long motivated a line of research into more dynamic routing strategies. One popular approach replaces the fixed K with a probability threshold (Huang et al., 2024; Yang et al., 2024), where the number of activated experts is determined by a cumulative probability score. Guo et al. (2025) and Yuan et al. (2025) make models learn to dynamically assign more experts to critical tokens. From a system perspective, NetMoE (Liu et al., 2025) explores dynamism by adjusting token allocation to enhance communication efficiency. Despite these varied explorations into dynamism, their primary focus is typically to discover a more optimal routing

policy to improve overall performance or efficiency. The resulting models are typically still deployed with a fixed inference behavior as they are not explicitly trained to accommodate different expert counts. The key requirement for elastic inference remains largely overlooked.

## 5.2 MATRYOSHKA REPRESENTATION LEARNING

Matryoshka Representation Learning (MRL) (Kusupati et al., 2022) is a framework for creating a single, adaptable representation where nested, truncated subspaces remain effective. It has been successfully applied in tasks like recommendation systems in both vision and language domains (Wang et al., 2024; Lai et al., 2024). The core principle has been extended from the output layer to internal model components (Devvrit et al., 2024). The principle's versatility is further demonstrated by its application in multimodal learning (Cai et al., 2025; Hu et al., 2024) and for addressing knowledge sharing challenges in federated learning (Yi et al., 2024).

## 6 DISCUSSION: MECHANISTIC PERSPECTIVE ON M-MOE

A central question in understanding the behavior of M-MoE is why the model preserves elasticity and does not collapse into the co-adaptive behavior commonly observed in standard static-$k$ MoE architectures. In this section, we provide an interpretation of the training dynamics that distinguishes M-MoE from standard MoE.

**Standard MoE Encourages Expert Co-Adaptation.** In a standard MoE model, all $k$ activated experts jointly optimize the loss at every training step. Since the model is trained exclusively on the aggregated output of the full $k$-expert ensemble, the experts naturally become mutually dependent. This co-adaptation intensifies as training progresses: the experts increasingly specialize in complementary micro-functions that are only optimal when combined with the others. As a result, the representation learned by any strict subset of experts is insufficient, and reducing $k$ during inference leads to notable performance degradation. This is an intrinsic tendency of static-$k$ MoE optimization.

**M-MoE Introduces a Hierarchical and Nested Inductive Bias.** In contrast, M-MoE explicitly trains nested subsets of experts. The smallest subset is required to independently optimize the objective, forming a stable core representation. Additional experts are trained to capture the residual error left by the smaller group. This design induces a hierarchical decomposition reminiscent of classical coarse-to-fine structures such as PCA components (Abdi & Williams, 2010) or nested subspace models (Rauba & van der Schaar, 2025). The optimization objective thus enforces a strict ordering of representational importance: the core subset must remain functional at all times, while additional experts provide incremental refinement. This structure prevents the collapse observed in standard MoEs and makes elasticity a stable, convergent behavior.

**Connections to Broader Matryoshka-Style Training.** This hierarchical residual-learning mechanism is consistent with phenomena observed in other Matryoshka-style training frameworks across different modalities and architectures Cai et al. (2025); Yi et al. (2024); Hu et al. (2024). These works independently show that nested-subset objectives naturally promote robust coarse-to-fine representations and prevent destructive co-adaptation during long training.

## 7 CONCLUSION

We introduce Matryoshka MoE, a simple yet powerful training framework that resolves the inherent brittleness of fixed-k MoE models and unlocks their potential for elastic inference. By training with a variable number of experts, we successfully build a coarse-to-fine hierarchy within the expert ensemble. The result is a single, versatile model capable of delivering performance comparable to a suite of specialist models. M-MoE's structural flexibility facilitates practical performance-efficiency trade-offs and opens new research into layer-wise, heterogeneous inference strategies.

## REPRODUCIBILITY STATEMENT

We are committed to ensuring the reproducibility of our work. The core principles and formulations of our M-MoE framework, including the key layer-wise training strategy, are described in the main text. A comprehensive account of our experimental setup is provided in Appendix B, which details the model architecture, data sources, training hyperparameters, parallelism configuration, and evaluation procedures. To further aid in implementation, Appendix C presents illustrative pseudo-code for the core M-MoE-layer logic, as well as for our custom analysis metrics: the Focused Spearman Correlation and Mean Off-Diagonal Similarity (MODS). Upon publication, we intend to release our source code and model checkpoints to allow for direct replication and to facilitate future research in this area.

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

## A  USE OF LARGE LANGUAGE MODELS

In preparing this manuscript, we utilized Large Language Models as an assistive tool. The LLM's role was confined to aiding in writing and implementation tasks. Specifically, it was used for language polishing, such as rephrasing sentences for clarity and correcting grammar. Additionally, it assisted in writing the Python script with `matplotlib` used for plotting experimental results and in formatting the LaTeX code for some tables. The human authors critically reviewed and edited all LLM-generated outputs, and retain full responsibility for the final content, methodology, and conclusions of this work.

## B  DETAILS OF EXPERIMENT SETUP

**Data.**   The model is trained on a diverse and high-quality dataset comprising a mixture of public and proprietary sources. This includes subsets of Numotron-CC (Su et al., 2025), deduped dclm (Li et al., 2024), deduped Fineweb-edu (Penedo et al., 2024), a large corpus of code, and synthetic data designed for reasoning tasks.

**Training.**   All experiments were conducted using the Megatron-LM (Shoeybi et al., 2019) on a cluster of NVIDIA A100 40G GPUs. The initial pre-training of our base model from scratch for 1 trillion tokens consumed approximately 180,000 GPU hours. Our main experiments, which involved the continual pre-training of our various M-MoE strategies and baselines for an additional 80B tokens (as detailed in Section 4.2), collectively consumed an additional 90,000 GPU hours. We trained the model with a sequence length of 4096, a global batch size of 16 million tokens, and a micro-batch size of a single sequence (4096 tokens). To efficiently scale training, we employed a hybrid parallelism strategy combining a pipeline-model-parallel-size of 2, a context-parallel-size of 2, and an expert-model-parallel-size of 8, with a tensor-model-parallel-size of 1. We used the AdamW optimizer with a weight decay of 0.1, $\beta_1 = 0.9$, $\beta_2 = 0.95$, and an epsilon of $10^{-9}$. The learning rate followed a Weighted Sample Decay (WSD) schedule with a linear decay profile, reaching a peak of 2.6e-4 after a 2,000-step warmup.

**Evaluation.**   We evaluate all models using `lm-evaluation-harness` (Gao et al., 2024). Performance is measured on a suite of common sense and knowledge-intensive benchmarks, including MMLU (Hendrycks et al., 2021), ARC Challenge (Clark et al., 2018), BoolQ (Clark et al., 2019), HellaSwag (Zellers et al., 2019), LogiQA (Liu et al., 2020), OpenBookQA (Mihaylov et al., 2018), and Winogrande (Sakaguchi et al., 2020). All evaluations are performed in a 5-shot setting.

## C  REPRODUCIBILITY DETAILS

To enhance the reproducibility of our key findings, this section provides illustrative pseudo-code for the core components of our methodology.

### C.1  LAYER-WISE M-MOE

As described in Section 3.2, the layer-wise M-MoE strategy introduces stochasticity at each layer. The pseudo-code below illustrates the core implementation for the **uniform sampling** variant, where a new number of experts, `k`, is sampled at the beginning of each router's forward pass.

```python
def forward(self, input):
    # Sample a new k for every forward pass
    self.topk = random.randint(self.k_min, self.k_max)

    # ... rest of standard MoE forward pass ...
```

Listing 1: Core logic for the M-MoE-layer router.

## C.2 FOCUSED SPEARMAN CORRELATION

In our analysis of Matryoshka Routing (Section 4.5), we introduced the Focused Spearman Correlation metric to quantify the consistency of the router's expert ranking. The core logic, shown for a single token, is provided below. It takes as input two distinct logit vectors—one from an inference run with $k_{large}$ and another with $k_{small}$—and computes the rank correlation on their union of selected experts.

```python
def get_focused_correlation(
    logits_large, logits_small, k_large, k_small):
    # Get top experts from each respective logit vector
    indices_large = torch.topk(logits_large, k_large).indices
    indices_small = torch.topk(logits_small, k_small).indices

    # Find the union of relevant experts
    relevant_indices = sorted(list(
        set(indices_large.tolist()) |
        set(indices_small.tolist())
    ))

    # Correlate scores from their original logit vectors
    scores_large = logits_large[relevant_indices]
    scores_small = logits_small[relevant_indices]
    corr, _ = spearmanr(scores_large, scores_small)

    return corr
```

Listing 2: Calculating Focused Spearman Correlation for one token.

## C.3 MEAN OFF-DIAGONAL SIMILARITY

To quantify expert specialization, as discussed in Section 4.6, we use the Mean Off-Diagonal Similarity (MODS) metric. A lower MODS score, indicating higher orthogonality among router weight vectors, signifies greater specialization. The pseudo-code below outlines the calculation.

```python
def calculate_mods(gate_weights):
    # L2-normalize each expert's weight vector
    normalized_weights = F.normalize(gate_weights, p=2, dim=1)

    # Compute the cosine similarity matrix
    sim_matrix = torch.matmul(
        normalized_weights, normalized_weights.T
    )

    # Mask the diagonal and average the rest
    num_experts = gate_weights.shape[0]
    mask = 1 - torch.eye(num_experts)
    off_diagonal_abs_sum = (sim_matrix * mask).abs().sum()
    mods = off_diagonal_abs_sum / (num_experts * (num_experts - 1))

    return mods
```

Listing 3: Core logic for calculating MODS.

# D ADDITIONAL EXPERIMENTS

## D.1 IMPROVING THROUGHPUT WITH AN ACTIVATION BUDGET

A practical challenge in implementing the Layer-wise M-MoE strategy is the volatility of computational cost. Because each layer independently samples its expert count, the total number of activated experts per token becomes a random variable, complicating memory provisioning and potentially hindering training throughput.

To mitigate this, we introduce an optional Activation Budget mechanism. This approach caps the total number of activated experts per token to a fixed budget, $B$. If the initial random sampling across layers exceeds this budget, we proportionally scale down each layer's expert count and then stochastically redistribute the remaining surplus slots until the budget is met exactly. This preserves layer-wise diversity while ensuring a predictable memory footprint.

We applied this mechanism to train a uniform sampling M-MoE-layer model with an average budget of 4.5 experts per layer. The results were highly effective: this budget-aware approach reduced peak GPU memory consumption by 10% compared to the unconstrained layer-wise training. This memory saving allow us to increase the micro-batch size from 3 to 4 on our hardware setup, resulting in an 8% improvement in overall training throughput.

As shown in Table 4, this significant gain in training efficiency was achieved with minimal impact on model performance. The budget-constrained model remains highly competitive with the unconstrained baseline, even outperforming it on average at lower inference expert counts ($k = 1$), confirming this technique as a valuable practical optimization for training elastic MoE models.

Table 4: Performance of the budget-constrained M-MoE-layer model (Avg. k=4.5). The final column shows the average score and, in parentheses, its difference from the unconstrained M-MoE-layer model in Table 1.

| Inf. k | MMLU | ARC-C | BoolQ | HellaS. | LogiQA | OBQA | WinoG. | Avg |
|---|---|---|---|---|---|---|---|---|
| 1 | 51.67 | 51.96 | 71.07 | 69.92 | 31.64 | 40.20 | 68.27 | 54.96 (+0.35) |
| 2 | 53.01 | 52.30 | 71.62 | 72.15 | 31.34 | 41.40 | 69.14 | 55.85 (-0.26) |
| 4 | 53.89 | 55.83 | 71.55 | 73.27 | 30.57 | 44.20 | 68.59 | 56.84 (+0.15) |
| 6 | 53.71 | 55.58 | 72.84 | 71.78 | 30.41 | 43.60 | 69.30 | 56.75 (+0.03) |

## D.2 IMPACT OF LONGER CONTINUAL PRE-TRAINING

To investigate how our proposed M-MoE framework scales with additional training, we extended the continual pre-training of the best-performing **M-MoE-layer** model. Starting from the 80B token checkpoint in Section 4.2, we trained the model for an additional 128B tokens, for a total of 208B tokens of continual pre-training.

The training dynamics are visualized in Figure 4. At the start (0 steps), the base model, which was only trained with k=1, shows a large performance gap when evaluated with more experts. The M-MoE training rapidly closes this gap; within the first 1,000 steps (16B tokens), the performance curves for all k values converge and begin to improve in unison. This visually confirms the effectiveness of our method in quickly instilling the Matryoshka property.

The final results at the 208B token checkpoint, presented in Table 5, show that the model's performance continues to improve consistently across all inference configurations compared to the 80B checkpoint. This indicates that the model had not yet reached saturation and benefits from further training. This experiment further validates that the M-MoE strategy is a scalable and robust approach that continually benefits from more training data and compute.

Table 5: Performance of the M-MoE-layer model after 208B tokens of continual pre-training. The final column shows the average score.

| Inf. k | MMLU | ARC-C | BoolQ | HellaS. | LogiQA | OBQA | WinoG. | Avg |
|---|---|---|---|---|---|---|---|---|
| 1 | 52.63 | 51.62 | 70.31 | 70.41 | 31.34 | 41.60 | 67.72 | 55.09 |
| 2 | 53.89 | 54.18 | 71.22 | 72.85 | 31.03 | 43.60 | 68.51 | 56.47 |
| 4 | 55.03 | 54.52 | 72.20 | 72.62 | 31.18 | 44.80 | 70.24 | 57.23 |
| 6 | 54.59 | 54.78 | 72.91 | 72.05 | 31.34 | 44.40 | 70.32 | 57.20 |

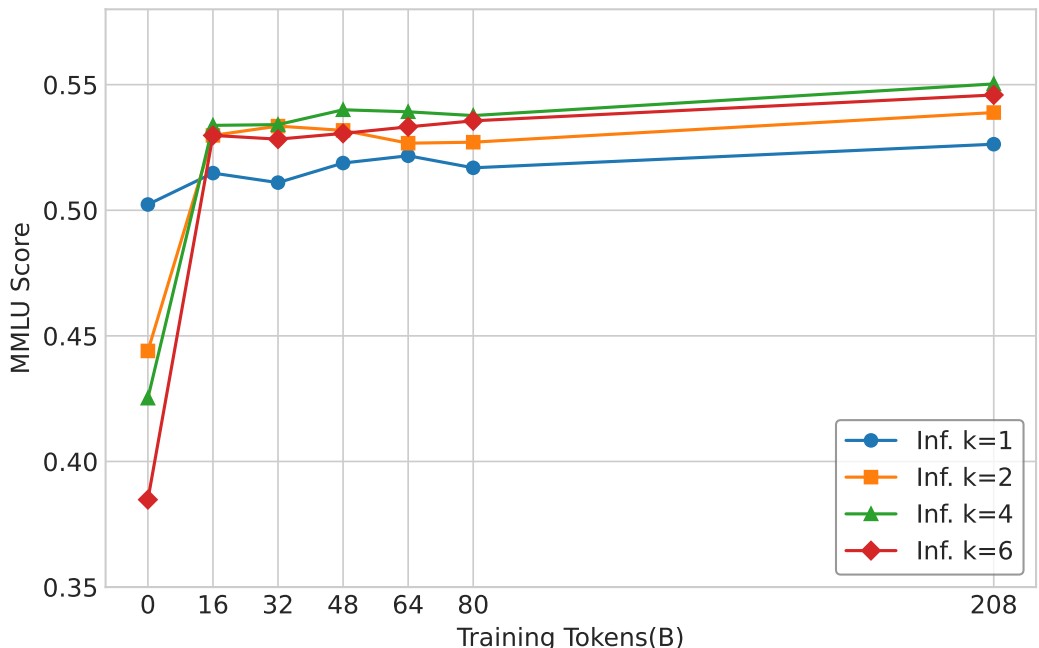

Figure 4: MMLU score of the M-MoE-layer model evaluated at different inference expert counts (k=1, 2, 4, 6) throughout continual pre-training. The x-axis represents training steps from the start of M-MoE training.

# E    LOSS CURVE IN FROM-SCRATCH PRE-TRAINING

Since benchmark scores can be unstable during the early stages of pre-training from scratch, we provide the training loss curves in Figure 5. As can be observed, the M-MoE model demonstrates training stability and performance that is fully comparable to the Top-k specialist baselines.

# F    LOAD BALANCE ANALYSIS

To address the concern regarding load balancing and the potential for a "routing shortcut" in our M-MoE model, we present a detailed analysis of its behavior during training. We compare our M-MoE-layer model against the Top-4 specialist baseline, focusing on the continual learning phase of our experiments. Our analysis utilizes two key metrics: the auxiliary load balancing loss and the direct distribution of tokens per expert.

Figure 6 shows the load balancing loss for both models. While the M-MoE-layer exhibits a brief period of temporary imbalance during the initial phase of continual training, it rapidly stabilizes. Subsequently, its load balancing loss becomes stable, indicating that the model achieves and maintains a stable and effective load distribution throughout the remainder of training.

For a more direct assessment, Figure 7 visualizes the token distribution across experts. The shaded areas represent the range between the minimum and maximum number of tokens assigned to any single expert within a global batch. The dashed horizontal lines indicate the theoretical average number of tokens per expert under a perfectly uniform load distribution. This value represents the ideal scenario where every expert receives an identical number of tokens. It is calculated by dividing the total token assignments in a global batch (global batch size $\times$ experts per token) by the total number of experts. Notably, for the M-MoE-layer model which randomly routes each token to $k$ experts where $k \in \{1, \ldots, 6\}$, the average number of experts per token is 3.5. This setting is comparable to the Top-4 baseline, resulting in similar theoretical average token loads as shown in Figure 7.

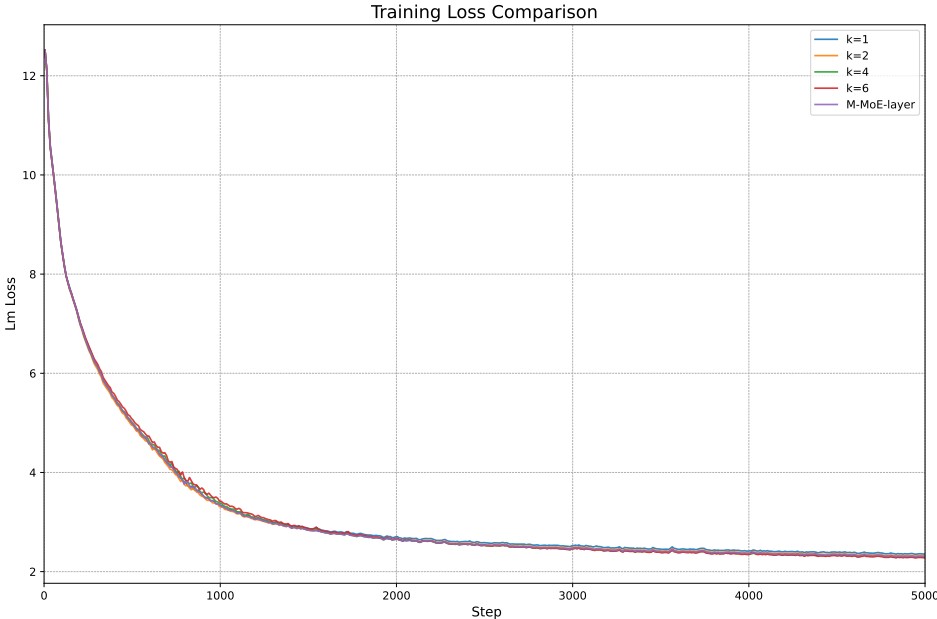

Figure 5: Comparison of training loss curves for our M-MoE-layer model and several Top-k specialist baselines (k=1, 2, 4, 6) during 80B token pretraining. All models were trained from scratch using identical data and configurations.

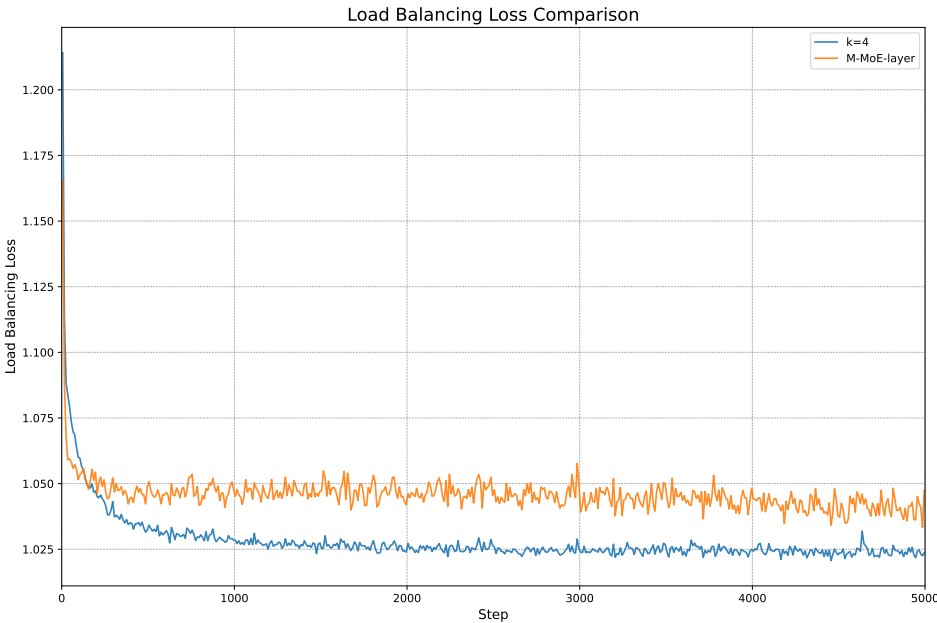

Figure 6: Comparison of the auxiliary load balancing loss for the M-MoE-layer and the Top-4 specialist baseline during continual training. The M-MoE model shows rapid stabilization after an initial adaptation phase.

The visualization demonstrates a clear and important trend for both models: the shaded regions progressively narrow and converge toward their respective average lines as training advances. This convergence provides strong evidence against the routing shortcut hypothesis. It confirms that no single expert is consistently favored (as the maximum value decreases) or starved (as the minimum value increases), and the token load becomes increasingly equitable across all experts over time. This affirms that the routing mechanism in M-MoE learns to distribute the load effectively without creating detrimental shortcuts.

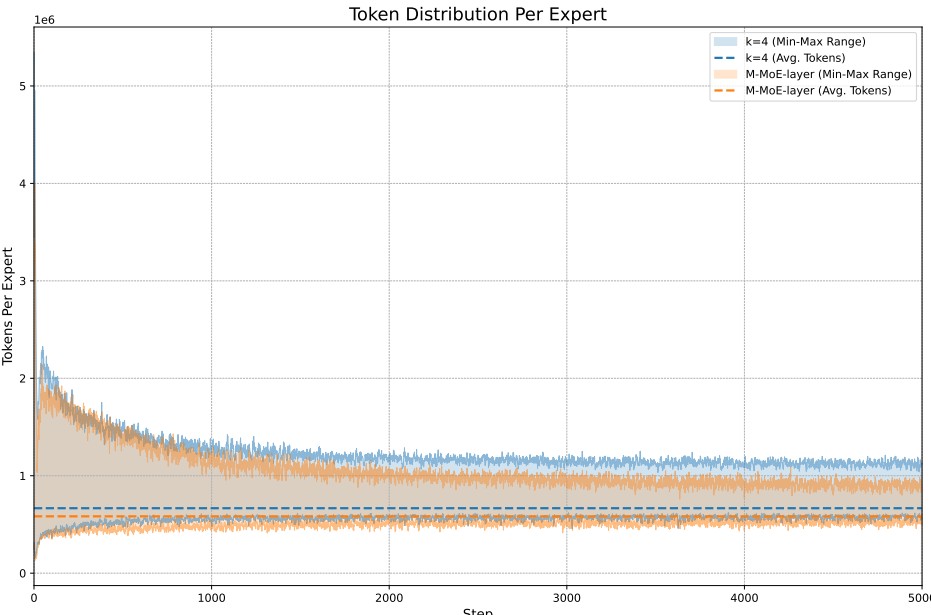

Figure 7: Token distribution per expert for the M-MoE-layer and Top-4 baseline. The shaded areas show the range from the minimum to the maximum number of tokens assigned to any expert. The dashed lines represent the theoretical average under perfect load balance. The narrowing of the shaded regions indicates that token distribution becomes more equitable as training progresses.

