# OpenReview forum: "Training Matryoshka Mixture-of-Experts for Elastic Inference-Time Expert Utilization"
_ICLR.cc/2026/Conference — ICLR 2026 Conference Withdrawn Submission_

### Official Review · Reviewer_tUR1 · 2025-10-27

**Soundness:** 1
**Presentation:** 2
**Contribution:** 2
**Rating:** 2
**Confidence:** 4

**Summary:**

This paper proposes Matryoshka MoE, that activates different number of experts per layer. This elastic activation can fit different computational budgets.

**Strengths:**

The exploration of M-MoE variants (such as batch-wise, layer-wise, etc) is comprehensive.

The writting is clear.

**Weaknesses:**

The biggest weakness of the work is its impracticality. Varying activation levels across layers introduce notable complexity for practical deployment, kernel implementation, and communication engineering. First, the model must be deployed on devices that can accommodate the maximum memory and computational requirements, right? However, when some layers activate fewer experts, this leads to more pipeline bubbles and results in resource waste. If the top-k activation is too large to fit on a device, your results indicate that continual training a specialist model with a smaller k performs well enough, and is more practical and cost-effective.

Additionally, the technical contribution is somewhat weak, as the proposed method is relatively straightforward.

Below are some weaknesses related to the method and its results.

1. Figure 1 is misleading because its x-axis omits the range from 4 to 7. I conducted some experiments on OLMOE (pre-trained with topk=8) by adjusting k during inference. The 0-shot results are shown below:
|k|arc-c|arc-e|piqa|hellaswag|winogrande|
|-|-|-|-|-|-|
|8|47.1|78.16|80.14|58.01|68.67|
|7|45.9|78.37|79.87|58.08|68.11|
|5|44.97|76.94|79.16|57.14|66.22|

When using an inference k smaller than the pre-trained k=8, performance does drop—but the decline is far less drastic than Figure 1 suggests. This discrepancy arises because the figure omits the x-axis range where the slight drop occurs. I recommend the authors include the full x-axis and corresponding experimental results to avoid overstating the severity of the issue illustrated.

2. The experiment setup in Table 1 is unrealistic and unconvincing. The base model for continual training only activates 1 expert per layer (topk=1), but pre-training MoE models with topk=1 is rarely practiced in real-world scenarios. This setup makes increasing k during continual training easy: the model could simply add some useless but harmless experts, while the single expert trained during pre-training remains the primary contributor to performance. How to rule out this possibility?

3. For the specialist baseline (k*=6) in Table 2, the authors should report results for Inf.k = 2, 3, 4. As noted earlier, performance drops only slightly when Inf.k is marginally smaller than the pre-trained k. Currently, Table 2 only includes Inf.k=1 and the pre-trained k, omitting intermediate values. This incomplete comparison might overstates M-MoE’s advantage.

4. Table 3 has confusing bold text and lacks significance tests. Is the value 55.42 incorrectly bolded? Moreover, I do not think there are statistically significant differences in the results in Table 3. It is hard to support the claim that "earlier layers are more critical."

5. If you consider a low MODS to be a good metric and believe that earlier layers are more critical, in Figure 3, the Matroshka model performs worse than the Top-K model in shallow layers due to its higher MODS scores. How can this be explained?

**Questions:**

1. What is your base model? Is it an open or private model? I did not find detailed descriptions of this model. Additionally, why was this model pre-trained with only one expert activated per layer? This is not a commonly used setup.

2. What is the formulation of the Focused Spearman Correlation?

3. See Weaknesses.

---

> ### Author Response · Authors · 2025-11-20
>
> We sincerely thank you for your thorough and constructive feedback. Your detailed comments have been invaluable in helping us improve the clarity and rigor of our work. We would like to address your concerns below.
>
> ### Regarding the Main Weakness: Impracticality
>
> We believe there is a misunderstanding regarding our primary intended use case and we would like to clarify with the following points:
>
> 1.  **Training vs. Inference Strategy:** A key point of clarification is that the layer-wise activation strategy you mentioned in the summary and weakness analysis is primarily employed during the **training phase**. In our main experiments, the **inference scenario** involves applying the same number of experts (`k`) across all layers. This setup targets a practical and elastic scenario, conceptually similar to the widely adopted Matryoshka Representation Learning (MRL) for embeddings. We agree that unlocking the full potential of direct layer-wise inference is difficult without tailored hardware and scheduling algorithms. However, we believe our work provides a valuable theoretical upper bound and can inform future hardware and algorithm design. While M-MoE demonstrates the potential for layer-wise inference, our primary focus remains on the more immediately practical uniform-`k` elastic deployment.
>
> 2.  **Deployment:** You raised an important point about deployment challenges and potential resource waste. While this is a valid concern, it is less of an issue in large-scale commercial settings. In such environments, a single large language model is typically distributed across multiple devices (e.g., different layers and experts on different devices) and deployed as multiple instances. The system can dynamically serve a high volume of external requests. The elasticity of M-MoE becomes a significant advantage here, as intelligent scheduling can allocate resources dynamically to efficiently handle fluctuating demand and avoid the pipeline bubbles and resource waste associated with underutilized, fixed-capacity models.
>
> 3.  **Advantages over Specialist Models:** Compared to training a new suite of specialist models, M-MoE offers two primary advantages:
>     *   **Reduced Training Overhead:** A single model can be trained to adapt to various inference budgets, saving the substantial computational cost of training multiple specialist models.
>     *   **Simplified Deployment:** It reduces operational complexity. Only one model artifact needs to be deployed and maintained, regardless of how the distribution of user requests (and their corresponding expert count requirements) changes over time.
>
>
> ### Responses to Other Weaknesses
>
> **W1: Figure 1's x-axis omits the range from 4 to 7.**
>
> Thank you for pointing this out. You are correct that for small deviations in `k`, the performance drop is modest, the degradation progressively accelerates as the inference `k` differs further from the training `k`. We have updated the figure in the paper to include the full range of x-axis values to present a more complete picture, and we have also revised the corresponding text description to accurately reflect this behavior. To provide a quick reference here, the MMLU scores for the `Qwen3-30B-A3B` model at inference `k` from 1 to 8 are:
>
> | k | 1 | 2 | 3 | 4 | 5 | 6 | 7 | 8 |
> |---|---|---|---|---|---|---|---|---|
> | MMLU Score | 24.33 | 33.98 | 66.22 | 75.24 | 78.78 | 80.07 | 80.67 | 80.99 |
>
> **W2: The experiment setup in Table 1.**
>
> The choice of a `k=1` pre-trained model was based on its availability from a prior project. Our from-scratch pre-training experiments presented in Table 2 may address your concern about whether this setup artificially inflates our results ("the model could simply add some useless but harmless experts"). The results in Table 2, which start from a randomly initialized model, show a trend consistent with Table 1. This rules out the possibility that the benefit is an artifact of the pre-training activation count. We believe presenting both settings provides a more convincing and robust validation of our method.
>
> **W3: More results for the specialist baseline (k=6) in Table 2.**
>
> We have updated Table 2 in our revision to include the results for the specialist baseline (`k=6`) at these intermediate inference `k` values.  Thank you for helping us present a more complete and nuanced comparison. The results for the specialist baseline (k=6) with the added intermediate k values are as follows:
>
> | Inf. k | MMLU | ARC-C | BoolQ | HellaS. | LogiQA | OBQA | WinoG. | Avg |
> | :--- | :--- | :--- | :--- | :--- | :--- | :--- | :--- | :--- |
> | 1  | 25.94 | 25.77 | 48.29 | 42.89 | 27.19 | 28.80 | 53.59 | 36.07 |
> | 2  | 25.49 | 36.60 | 44.80 | 54.88 | 28.11 | 35.00 | 53.83 | 39.82 |
> | 4  | 29.00 | 38.74 | 49.39 | 56.90 | 28.26 | 35.20 | 55.17 | 41.81 |
> | 6  | 30.20 | 38.99 | 54.39 | 56.97 | 29.80 | 35.20 | 56.59 | 43.16 |```

---

> ### Author Response · Authors · 2025-11-20
>
> **W4: Bold text and statistically significant differences in Table 3.**
>
> 1. The bolding was intended to highlight the configuration with the **greatest performance change** relative to the baseline, not necessarily the absolute best score. For "Increasing Capacity," this corresponds to the largest performance gain, while for "Decreasing Capacity," it highlights the largest performance drop. We have added an explicit explanation to the paper's caption. While we acknowledge that performance shifts are subtle and some benchmarks are less sensitive, the trends in the MMLU and Avg columns are consistent: increasing capacity in earlier layers yields the largest benefit, and decreasing it in those same layers causes the most significant degradation. To further strengthen this point, we have conducted an additional experiment on DeepSeekV2-Lite which show a consistent conclusion in Appendix.
> 2. The relatively small differences in the final 'Avg' scores are due to the fact that some benchmarks are less sensitive to minor performance variations. We have added a note to this effect in our revision. We find that the MMLU score is the most reliable indicator of performance shifts in this context, as it is a widely-used, large-scale benchmark that provides stable metrics. To demonstrate the statistical significance of the MMLU results, we report the accuracy and standard error, calculated using the `lm-evaluation-harness` framework:
>
> | Group | Activation Pattern | Accuracy  | Stderr  | Change  |
> | :--- | :--- | :--- | :--- | :--- |
> | **Baseline** | `[2, 2, 2, 2]` | 52.71 | 0.421 | - |
> | **Increasing Capacity** | `[3, 3, 2, 2]` | 53.35 | 0.422 | **+0.64** |
> | | `[2, 3, 3, 2]` | 53.13 | 0.422 | +0.42 |
> | | `[2, 2, 3, 3]` | 52.90 | 0.422 | +0.19 |
> | **Decreasing Capacity** | `[1, 1, 2, 2]` | 51.63 | 0.422 | **-1.08** |
> | | `[2, 1, 1, 2]` | 52.56 | 0.422 | -0.15 |
> | | `[2, 2, 1, 1]` | 52.78 | 0.421 | +0.07 |
>
> **W5: MODS across layers.**
>
> This is a very insightful question. The MODS metric reflects the degree of expert specialization. It is expected that different layers *within a single model* will exhibit varying levels of specialization, as they likely perform different functions (e.g., more general tasks in early layers, more specialized tasks in middle layers). Therefore, comparing the MODS of a shallow layer to a deep layer within the Matryoshka model does not indicate whether one layer is "more critical" or "performing worse."
>
> Our claim is based on the **inter-model comparison**: for corresponding layers, the Matryoshka model consistently shows a lower MODS score than the standard Top-k model across nearly the entire network depth, as seen in Figure 3. This demonstrates that our M-MoE training method successfully induces greater expert specialization overall.
>
> ### Responses to Questions
>
> **Q1: Base model.**
>
> It is a private model developed as part of a separate project focused on training sparse models. During that project, we found this model to be stable and performant in both training and inference. The primary advantage of using this model over a public one was to ensure complete consistency in data distribution and training configurations between pre-training phase and our continual pre-training experiments. This helps isolate the effects of our M-MoE strategy by minimizing performance variance that could arise from domain shifts in data. Given the prohibitive cost of pre-training a new model from scratch, we opted to build upon this existing checkpoint. Our checkpoints will be publicly available.
>
> **Q2: Focused Spearman Correlation.**
>
> We provided the code for the Focused Spearman Correlation in Appendix C.2 of our paper.
>
> We sincerely thank you once again for your  constructive feedback. We hope that the clarifications and additional experiments provided  address your concerns.

---

> > ### Comment · Reviewer_tUR1 · 2025-11-22
> > **Thank you for your detailed response. Some of my concerns are addressed while many are not.**
> >
> > 1. In the industrial expert parallel scenario you mentioned where your method is claimed to be useful, please explain how you determine the value of k when a batch contains queries from different users—some difficult and some simple. How is this decided automatically, and can k vary per sample? Also, please do not claim, "we believe our work provides a valuable theoretical upper bound and can inform future hardware and algorithm design." Your paper contains no theoretical proofs.
> >
> > 2. Why are the results for top-p and M-MoE-global-batch in Table 1 exactly the same?
> >
> > 3. After seeing the results of from-scratch pre-training, my concerns were intensified for following reasons:
> >
> >     (i) In Table 2, the activated experts 2, 4, and 6 for your method are 42.63, 42.67, and 42.60, respectively. Doesn’t this reinforce my doubt that "only some experts are useful in this model, while the others are trained to be useless but harmless"?
> >
> >     (ii) Your method, even with 6 out of 6 experts activated, does not outperform the specialist baselines (native 6 activated out of 6 experts), perhaps due to more experts are trained to be useless but harmless.
> >
> >     (iii) In Table 2, your method only improves by a maximum of 0.12% compared to a native 2 out of 2 activated model (42.67 - 42.55), which is not significant. Based on my experience, with only 80B tokens, fluctuations between different checkpoints are far greater than (0.12 / 42.55).
> >
> >     (iv) Since you stated that MMLU is the most reliable benchmark among all your evaluations, in Table 2, your method (6 out of 6, 4 out of 6, and 2 out of 6) shows almost no difference compared to the native 6, 4, and 2 expert models.
> >
> > 4. Regarding your private model, my concerns remain. The commitment to open-sourcing is commendable, and I will accept your experimental setup *for the subsequent discussion.* However, I have to note that a 1-in-96 MoE behaves substantially differently from high-sparsity models like an 8-in-256 MoE. Their flexibilities differ by orders of magnitude (C(256,8) vs. 96). I acknowledge that retraining a base model was infeasible, so I note this issue for the convenience of the Area Chairs in subsequent discussions.

---

> > > ### Author Response · Authors · 2025-11-25
> > >
> > > Thank you for your prompt response. We appreciate the opportunity to continue this discussion and address your remaining concerns.
> > >
> > > **Q1: Application Scenario**
> > >
> > > (1) We would like to further clarify the intended application scenario. In our framework, the value of $k$ is not determined automatically by the model, but is rather specified by the user based on price or latency constraints. This aligns with the standard setting of Matryoshka Representation Learning (MRL) [1][2], where the dimension size is chosen based on downstream storage or efficiency requirements. For example, a service provider could use a single M-MoE model to power multiple API tiers without the need to deploy and maintain separate models. Users can select the budget based on their specific price and latency requirements.
> > >
> > >  While dynamically predicting the optimal $k$ per sample is an interesting research direction, our work focuses on the fundamental capability of the model to adapt to varying specified budget, rather than the policy for selecting that budget.
> > >
> > > (2) We agree that the phrase "theoretical upper bound" was imprecise. We intended to convey that in an ideal scenario with optimal scheduling algorithms, pipeline bubbles and resource waste could be minimized, thereby reaching a resource utilization ceiling. We believe that "targeting higher resource utilization efficiency" is a more appropriate expression than "theoretical upper bound" in this context.
> > >
> > > **Q2: Identical Results in Table 1**
> > >
> > > We apologize for this confusion. This was a copy-paste error where the LaTeX code for the `M-MoE-global-batch` row was used as a placeholder for `Top-p` during formatting, but the values were not updated. This error has now been corrected in the revised manuscript. Thank you for pointing this out.
> > >
> > > **Q3: Analysis of From-Scratch Pre-training Results**
> > >
> > > **(i) Regarding the flat performance across k:**
> > > We provide the following explanation:
> > > (a)The observation that performance is relatively flat across $k=2, 4, 6$ in Table 2 is primarily due to the early stage of pre-training. At this stage, the model is far from reaching its capacity limit. The training dynamics naturally favor the top-ranked experts (greedy optimization), making it difficult for lower-ranked experts to play a role.
> > >
> > > Importantly, this trend is also observed in the specialist baselines. As shown in Table 2, the specialist models trained with fixed $k=2, 4, 6$ achieve MMLU scores of 30.40, 30.96, and 30.20, and Average scores of 42.55, 42.62, and 43.16, respectively. The flatness therefore appears to be a common phenomenon at this training stage and data scale.
> > >
> > > (b) We respectfully disagree with the interpretation that additional experts are useless. Instead, we observe that as training continues and the optimization efficiency of the dominant experts decreases, the model increasingly leverages lower-ranked experts as a more effective direction for further improvement. We extended the training of our from-scratch M-MoE model for additional 80B tokens. The results below show that lower-ranked experts are increasingly contributing to performance:
> > >
> > > | Inf. k | **MMLU** | ARC-C | BoolQ | HellaS. | LogiQA | OBQA | WinoG. | **Avg** |
> > > | :--- | :--- | :--- | :--- | :--- | :--- | :--- | :--- | :--- |
> > > | 1 | **32.94** | 43.34 | 62.51 | 59.60 | 28.88 | 34.40 | 56.83 | **45.50** |
> > > | 2 | **34.89** | 44.20 | 57.16 | 63.01 | 29.49 | 37.60 | 57.46 | **46.26** |
> > > | 4 | **35.24** | 44.37 | 60.31 | 63.12 | 28.73 | 37.60 | 58.56 | **46.85** |
> > > | 6 | **34.93** | 44.03 | 62.26 | 62.56 | 28.42 | 37.40 | 59.19 | **46.97** |
> > >
> > > **(ii) & (iii) & (iv) Performance Parity**
> > > We would like to clarify a misunderstanding regarding the goal of M-MoE. Our objective is not to significantly outperform the specialist baselines at every width, but to achieve comparable performance to a suite of specialist models using a single model [1][3].
> > >
> > > The value proposition of M-MoE is elasticity and efficiency:
> > > *   Training: We obtain functionality equivalent to multiple models for the training cost of one.
> > > *   Deployment: We significantly reduce deployment complexity by maintaining a single artifact.
> > >
> > > Therefore, the fact that M-MoE achieves parity (or very slight improvements) with specialist baselines is a success, not a failure. It demonstrates that we can compress the capabilities of multiple fixed-$k$ models into one elastic architecture without significant performance loss.
> > >
> > > **Q4: Model Sparsity**
> > >
> > > Thank you for your understanding regarding the experimental costs. Given our compute constraints, we have invested as much training as possible. After training, the flexibility of M-MoE expands to C(6,96).
> > >
> > > **References:**
> > >
> > > [1] A. Kusupati, et al. Matryoshka Representation Learning. NeurIPS 2022.
> > >
> > > [2] https://zilliz.com/blog/matryoshka-representation-learning-method-behind-openai-text-embeddings
> > >
> > > [3] Y. Wang, et al. Train Once, Deploy Anywhere: Matryoshka Representation Learning for Multimodal Recommendation. EMNLP 2024.

---

> > > > ### Comment · Reviewer_tUR1 · 2025-11-25
> > > >
> > > > 1. The application scenario now makes sense.
> > > >
> > > > 2. Your results show that lower-ranked experts are increasingly contributing to performance in your model, with additional 80B tokens. However, in a from-scratch pre-training scenario over trillions of tokens, how can you prove that your method will not ultimately converge to the behavior of a standard (static-k) MoE (i.e., low-ranked experts are also crucial, as demonstrated by the significant performance drop when k is reduced).
> > > >
> > > >     I realized that a fundamental problem in our discussion is: what constitutes a reasonable performance drop when using a smaller k? I believe it is likely difficult to quantify. I do not think that simply training the specialist for another 80B tokens and reporting results would resolve this, as the interpretation can be highly subjective: If the specialist's performance drops more, one could interpret it as "the baseline's low-ranked experts are less useless and harmless" (a negative for your model) or as "continued training reveals the vanilla MoE's inability to support dynamic k inference" (a negative for the baseline). If the drop is similar to your method's, it would be inconclusive, requiring even more training. If the specialist's drop is smaller, it could be interpreted either as a positive for the baseline ("the baseline better supports dynamic k") or a negative ("the baseline's experts are more useless but harmless").
> > > >
> > > >     Through our discussions, I found that there needs to be an objective, mechanistic analysis to prevent such subjective interpretations of the experimental results. Downstream performance metrics alone cannot resolve our fundamental disagreement on the method's effectiveness, given our divergent interpretations of these scores.
> > > >
> > > > Nevertheless, I appreciate the authors' efforts during the rebuttal period, and I will raise my rating to 4.

---

> > > > > ### Author Response · Authors · 2025-11-25
> > > > >
> > > > > We sincerely thank you for your meaningful feedback and the engaging discussion. We appreciate the valuable perspectives raised during this process—particularly regarding the interpretation of expert utilization. We will incorporate these insights into the Discussion section of our final manuscript. This has been a highly constructive dialogue. Thank you once again.

---

> ### Author Response · Authors · 2025-11-30
>
> We sincerely thank the reviewer for the constructive exchange and for raising the rating to 4. We have addressed most concerns with clarifying the deployment scenario, improving the presentation and including more results.
>
> The remaining point raised by the reviewer is the desire for a mechanistic, objective explanation beyond downstream results to explain why M-MoE maintains elasticity and does not collapse into the behavior of a standard static-k MoE.
>
> To address this, we have added a discussion section in the revised manuscript that provides the following conceptual clarification:
>
> - **Standard MoE encourages expert co-adaptation.**
> In standard MoE training, all k activated experts jointly optimize the loss for every forward pass. Because the model is optimized only for the aggregated output of the full k-expert ensemble, this setup naturally leads to co-adaptation: experts become mutually dependent and tightly coupled. As training continues, this dependence deepens, which explains why reducing k at inference causes performance degradation.
>
> - **M-MoE imposes a strict hierarchical inductive bias.**
> Because M-MoE trains nested expert subsets, the smallest subset must independently optimize the objective, while additional experts learn only the residual error left by the smaller group.
> This leads to a decomposition structure similar to PCA or nested subspace models， where higher-ranked components refine but do not overwrite the performance of lower-ranked components.
> Therefore, elasticity is a stable convergent behavior enforced by the optimization objective.
>
> To further contextualize this explanation, we note that similar hierarchical and residual-learning mechanisms have independently emerged in other Matryoshka-style training frameworks across different modalities and architectures [1,2,3]. These works also observe that nested-subset objectives naturally encourage stable, coarse-to-fine representations, suggesting that this structural inductive bias is a general consequence of the Matryoshka training paradigm rather than a phenomenon unique to our model.
>
>
> We thank the AC and reviewer again for their careful evaluation.
>
> **References:**
>
> [1] W. Hu, et al. Matryoshka query transformer for large vision-language models. NeurIPS 2024.
>
> [2] L. Yi, et al. Federated model heterogeneous matryoshka representation learning. NeurIPS 2024.
>
> [3] M. Cai, et al. Matryoshka multimodal models. ICLR 2025

---

### Official Review · Reviewer_g7C7 · 2025-10-28

**Soundness:** 3
**Presentation:** 3
**Contribution:** 3
**Rating:** 4
**Confidence:** 4

**Summary:**

This paper introduces Matryoshka-MoE (M-MoE), a training pipeline designed to enable elastic MoE inference.
The authors argue that fixed-*k* training limits the flexibility of MoE inference.
A reduction in activated experts will lead to significant performance degradation in a well-trained MoE.
The design of M-MoE is straightforward, involving the activation of varing numbers of experts during training.
M-MoE achieves competitive results across the full suite of specialist models under different activation setups.
The authors propose different M-MoE strategies and demonstrate that their layer-wise M-MoE is particularly effective.
Extensive experiments are presented to support the authors’ claims and highlight the effectiveness of M-MoE.

**Strengths:**

- The research question addressed in this paper is interesting and meaningful, providing researchers with insights into advancing MoE inference efficiency.

- The idea of introducing Matryoshka routing into MoE training is straightforward and well-motivated.

- The authors perform extensive experiments to demonstrate the effectiveness of their M-MoE, including continual pre-training with 1T tokens for a 20B language model and from-scratch pre-training, which strengthen their claims.

**Weaknesses:**

- I have concerns regarding some of the experimental results, as they do not adequately support the authors' claims or demonstrate the effectiveness of M-MoE (See Q1 & Q2).

- The authors lack empirical evidence to show that M-MoE can truly improve MoE inference efficiency, raising doubts about its practical applicability in real-world scenarios (See Q3 & Q4).

**Questions:**

---
Q1: What's the performance when number of Activated Experts is set between 3 and 8 in Figure 3.

I suggest the authors provide a finer-grained analysis of activated expert counts to better illustrate the trend and shape of the performance degradation.

---
Q2: Can the authors also inlude the top-k sepecialist baseline results beyond Inf. k = 1 and its native activation count in Table 1 and 2 ?
(e.g Top-k (k=6), *Inf.k* = 2,4)

I have the following concerns regarding Table 1 and 2.

(1) The authors state that the Top-k baseline suffers a severe performance drop when evaluated with a different number of activated experts.
How does an MoE trained with 6 experts perform when evaluated with 2 or 4 activated experts? Is its performance comparable to that of your M-MoE variants?

(2) The contitual pre-training setup in Table 1 seems unusual, where only a single expert is activated during per-training.
What is the performance of M-MoE when more experts are activated during the pre-training phase?

---
Q3: What's the end-to-end training throughput of your M-MoE variants and top-k specialist baselines?

What is the actual training efficiency of M-MoE relative to the baseline methods?

Does M-MoE introduce additional complexity to the pipeline, thereby reducing compatibility with existing MoE training and deployment frameworks?

Could the authors include a discussion of the following points:（1）training throughput and (2) inference speed comparisons, to provide further clarity.

---
Q4: What's the loading balance of M-MoE models?

I am curious whether Matryoshka routing introduces a shortcut in MoE routing — i.e., some experts are more easily activated and consistently assigned large routing weights, which could explain the observed performance improvement over Top-1.

A comparison of load balance across experts would help clarify any potential loading imbalances and shed light on the shortcut issue.

---

> ### Author Response · Authors · 2025-11-20
>
> We sincerely thank you for your thorough and constructive review. Your concerns regarding the experimental evidence and practical efficiency are very important, and we appreciate the opportunity to provide additional data and clarifications to address them.
>
> **Q1: Performance for Activated Experts between 3 and 8**
>
> Thank you for this suggestion. To provide the finer-grained analysis you requested, we have updated Figure 1 in the manuscript. The new figure includes more data points and better illustrates the performance trend. When the number of activated experts at inference deviates only slightly from the trained value (k=8), the performance degradation is gentle. The drop becomes much more significant as the deviation increases, which supports our claim about the brittleness of the fixed-k training paradigm.
>
> **Q2: Additional Top-k Specialist Baseline Results & Pre-training Setup**
>
> **(1) Performance of Specialist Models at Non-Native `k` Values**
>
> We have updated Tables 1 and 2 in the manuscript to include the performance of specialist models at more non-native `k` values. The updated results reveal a clear and consistent trend: At these non-native `k` values, the specialist models are consistently outperformed by our M-MoE. Furthermore, this performance gap widens as the inference `k` deviates further from the model's native activation count.
>
> **(2) Rationale for the Continual Pre-training Setup**
>
> We understand why our continual pre-training setup might seem unusual. This choice was a practical one, based on the availability of a robust base model. The Top-k=1 model we used was developed as part of another project focused on training sparse models. During that project, we found this model to be stable and performant in both training and inference.
>
> Given the prohibitive cost of pre-training a new model from scratch, we opted to build upon this existing checkpoint. To show the general applicability of our method, we also provide results from scratch in **Section 4.3**, which confirm the effectiveness of M-MoE.
>
> **Q3: Training/Inference Efficiency and Implementation Complexity**
>
> Thank you for raising these critical questions about practical applicability.
>
> **(1) Implementation Complexity**
>
> M-MoE does not introduce any complexity to the training pipeline. As we show with pseudo-code in Appendix C.1, the core mechanism can be implemented by adding just a few lines of code to randomize `k` during the forward pass. This does not create any compatibility issues with existing MoE training or deployment frameworks.
>
> **(2) Training Throughput**
>
> We have calculated the throughput in tokens per GPU per second. The results are as follows:
>
> | Activation Strategy | Throughput (tokens/gpu/s) |
> | :--- | :--- |
> | Top-k (k=1) | 4076.4 |
> | Top-k (k=2) | 3746.2 |
> | Top-k (k=4) | 3211.6 |
> | Top-k (k=6) | 2876.9 |
> | M-MoE (k ∈ [1, 6]) | 3230.2 |
>
> The throughput differences are not substantial. This is primarily because in large-scale distributed training, the communication overhead becomes a significant bottleneck, masking the smaller computational differences between these activation strategies.
>
> The primary efficiency gain of M-MoE is not in its raw training speed for a single run but in its overall training and deployment economy. With M-MoE, a single training run produces one versatile model that can be deployed across multiple inference budgets. This approach saves the immense costs associated with training and deploying a suite of separate specialist models.
>
> **(3) Inference Speed**
>
> In the primary use case for M-MoE—inference with a standard top-k routing strategy—the inference logic is identical to the baseline, and thus there is no extra overhead.
>
> To further demonstrate the predictable performance of different strategies that our model enables, we compare several fixed-k strategies with a layer-wise strategy where we activate 3 experts for even layers and 5 for odd layers.
>
> | Activation Strategy | Inference Speed (tokens/s) |
> | :--- | :--- |
> | k=1  | 9.0138 |
> | k=2  | 8.0934 |
> | k=3  | 7.3449 |
> | k=4  | 6.7213 |
> | k=5  | 6.1961 |
> | k=6  | 5.7472 |
> | [3,5,3,5,...] (avg k=4) | 6.6649 |

---

> ### Author Response · Authors · 2025-11-20
>
> **Q4: Load Balancing in M-MoE Models**
>
> Load balance during pre-training is a critical aspect. We maintained the use of a load balancing loss throughout our training process. We have monitored its behavior and found that while there is a temporary load imbalance in the very early stages of M-MoE training, the model quickly recovers and achieves a load balance state comparable to standard MoE training.
>
> Furthermore, we tracked the minimum and maximum number of tokens assigned to each expert. Our analysis shows that as training progresses, both of these values converge toward the mean, indicating that no single expert is being consistently favored or starved. This provides strong evidence that M-MoE does not create a detrimental routing shortcut. We add these results in Appendix F to show the load balance over time.
>
>
> We hope these additional results and clarifications address your concerns. We thank you again for your constructive and detailed feedback, which has helped us improve the clarity and completeness of our paper.

---

### Official Review · Reviewer_kZjX · 2025-10-31

**Soundness:** 2
**Presentation:** 2
**Contribution:** 2
**Rating:** 6
**Confidence:** 3

**Summary:**

This paper proposes an interesting setting for MoE models: the number of activated experts (K) is altered at inference time. The authors identify that the fixed Top-K training paradigm leads to severe performance degradation if K is changed, preventing the elastic trade-off between computational cost and model quality. Then, the author proposes some strategies to handle this and conducted corresponding experiments.

**Strengths:**

1. The author proposes a training framework to train a moe flexible to different K while inference, while achieving performance similar to top-K pretrained ones.
2. Many experiments have been done.
3. The elastic inference setting is interesting.

**Weaknesses:**

1. I’m not sure whether this method should be compared with dynamically-activated MoEs (i.e., those that allow the model to choose the computing budget at inference time, rather than training with top‑p but inferring with top‑k), or whether the authors want to emphasize the necessity of allowing humans, rather than the model, to choose the computing budget.

2. Could you also report the perplexity or the loss curve, since benchmark results may vary due to randomness (especially in the from-scratch setting)?

3. Have you tried continuing training the Top‑K model for a while? I mean, using the same training cost as M-MoE to train a Top‑K model, then applying a short period of continued training to obtain versions with different K values. (This is somewhat similar to your current setting, but I would prefer that the base model not be an MoE with top‑k = 1.)

**Questions:**

1. It is quite interesting that your M-MoE model achieves better performance than Top-k = 4 model with an average k < 4. Can you provide some insights or explanations?
2. I suggest focus on from-scratch setting more since the continue-training setting is a little wired (the base model is top-1 yet you continue it to increase topk)

---

> ### Author Response · Authors · 2025-11-20
>
> Thank you for your thoughtful feedback. We appreciate the opportunity to clarify our work and address the questions raised.
>
> **Regarding Questions:**
>
> **Q1: M-MoE achieving better performance than Top-k=4 with an average k < 4.**
>
> This is a very insightful observation, and we thank you for making it. We offer a potential explanation: The M-MoE training paradigm fosters greater expert specialization. As shown in our analysis in Section 4.6, M-MoE training results in lower MODS, indicating that experts become more distinct and less redundant. By forcing the model to perform well with varying numbers of experts, M-MoE introduces an implicit learning objective: top-ranked experts must capture coarse-grained, essential information, while lower-ranked experts must provide progressively finer-grained details. This structured division of labor may lead to a more effective and efficient use of the model's capacity. This view is supported by other works [1, 2, 3] which also suggest that enhancing expert specialization can lead to improved model quality.
>
> **Q2: Focusing more on the from-scratch setting.**
>
> Our choice for the continual training setting was primarily driven by cost considerations. Training a new base model from scratch requires a vast number of tokens to achieve stable performance, which is prohibitively expensive.
> The Top-k=1 model is a model developed as part of another project focused on training sparse MoE models. During that project, we found this model to be stable and performant in both training and inference.
> Given the prohibitive cost of pre-training new base models, we opted to build upon this existing checkpoint. However, we agree with you that the from-scratch results are important for demonstrating the fundamental benefits of our method. That is why we included the from-scratch experiments in Section 4.3, which confirm M-MoE's effectiveness.
>
> **Regarding Weaknesses:**
>
> **W1: Comparison with dynamically-activated MoEs.**
>
> As you mentioned, the goal of M-MoE is to empower humans (or deployment systems) to choose the computing budget at inference time. Our framework is designed to produce a single, versatile model that performs well across a range of pre-defined `k` values. This is distinct from the goal of the dynamic MoE methods, which aim to have the model learn a more optimal, per-token activation strategy. We discuss these methods in the related work to clarify the positioning of our work.
>
> **W2: Reporting loss curves.**
>
> Thank you for this valuable suggestion. We add loss curves in Appendix E in the revised manuscript. They demonstrate that M-MoE achieves comparable convergence and performance to the specialist baselines.
>
> **W3: Alternative training strategy (Continue training a strong Top-K model).**
>
> You propose an good alternative approach. However, due to the prohibitive computational costs of pre-training another base model from scratch, we did not explore this specific avenue. We believe that the evidence from our two primary settings—continual training on a Top-k=1 model and training entirely from scratch (Section 4.3)—already provides strong, complementary support for the effectiveness of the M-MoE framework.
>
>
>
> Thank you again for your constructive review. We believe that addressing these points will significantly strengthen the paper.
>
> **References:**
>
> [1] H. Guo, et al. Advancing Expert Specialization for Better MoE. neurips 2025.
>
> [2] A. Cheng, et al. ERMoE: Eigen-Reparameterized Mixture-of-Experts for Stable Routing and Interpretable Specialization. arxiv 2025.
>
> [3] L. Mirvakhabova, et al. Dirichlet-Prior Shaping: Guiding Expert Specialization in Upcycled MoEs. arXiv 2025.

---

### Official Review · Reviewer_WfL3 · 2025-10-31

**Soundness:** 3
**Presentation:** 3
**Contribution:** 3
**Rating:** 6
**Confidence:** 3

**Summary:**

This paper introduces Matryoshka Mixture-of-Experts (M-MoE), a training framework designed to enable elastic inference in sparse MoE language models. Standard Top-K routing results in sharp performance degradation when the number of activated experts is reduced at inference time. M-MoE mitigates this by randomizing the number of active experts (K) during training. The authors compare three levels of stochasticity — global batch, micro-batch, and layer-wise — and find that layer-wise performs best in achieving stable, elastic performance.
Experiments include continual pre-training from a 1T-token single-expert model (additional 80B and 208B tokens), showing that M-MoE maintains high accuracy across K=1–6.

**Strengths:**

1. Identifies a critical limitation of Top-K routing: sharp accuracy drop when reducing K.
2. Proposes a simple yet effective training method that yields elastic inference.
3. Demonstrates strong empirical validation across multiple training regimes (continual and from-scratch).
4. Provides clear pseudo-code and detailed experimental settings, supporting reproducibility.

**Weaknesses:**

1. Missing quantitative comparison of training throughput (FLOP/s, GPU utilization) between Top-K and M-MoE.
2. Conceptually incremental relative to previous Matryoshka and dynamic MoE literature.

**Questions:**

1. Could the authors provide explicit FLOP/s or wall-clock throughput comparisons with fixed-K training?

---

> ### Author Response · Authors · 2025-11-20
>
> We sincerely thank the reviewer for their valuable feedback. We would like to address the weaknesses and questions raised as follows.
>
> **1. Quantitative Comparison of Training Throughput**
>
> Thank you for this important question. We have calculated the training throughput in tokens per GPU per second, measured by global step time and the number of tokens processed. The results are as follows:
>
> | Activation Strategy | Throughput (tokens/gpu/s) |
> | :--- | :--- |
> | Top-k (k=1) | 4076.4 |
> | Top-k (k=2) | 3746.2 |
> | Top-k (k=4) | 3211.6 |
> | Top-k (k=6) | 2876.9 |
> | M-MoE (k ∈ [1, 6]) | 3195.2 |
>
> The throughput differences between the various activation strategies are not substantial. This is primarily because the communication overhead becomes a significant bottleneck in distributed training, masking the computational differences between these settings.
>
> The primary efficiency gain of M-MoE is not in its raw training speed for a single run but in its overall training and deployment economy. With M-MoE, a single training run produces one versatile model that can be deployed across multiple inference budgets. This approach saves the immense costs associated with training and deploying a suite of separate specialist models.
>
> **2. Relationship and Distinction from Matryoshka and Dynamic MoE Literature**
>
> We appreciate the reviewer's comment on the conceptual lineage of our work. We would like to clarify the key distinctions and novel contributions of our approach.
>
> *   **Matryoshka Representation Learning (MRL):** Our work is directly inspired by the core principle of MRL. While MRL instills a nested, coarse-to-fine structure within the *embedding dimension* of a representation vector, we successfully translate this idea to the architectural level of expert collaboration in Mixture-of-Experts models.
>
> *   **Dynamic MoE:** Our work targets a fundamentally different objective than the dynamic MoE methods. The goal of dynamic MoE is typically to have the model learn an activation strategy for different tokens. In contrast, our focus is on enabling elastic inference: creating a single model that can be intentionally deployed under various, pre-defined computational budgets.
>
> We hope these clarifications address your concerns. Thank you again for your thoughtful review.

---

### Author Response · Authors · 2025-11-27

Dear Reviewers,

We appreciate your valuable comments on our paper. We have carefully addressed the weaknesses and questions raised in our rebuttal.

As the discussion period is coming to an end, we would greatly appreciate it if you could spare a moment to check our responses. We hope that our clarifications and the additional results have addressed your concerns.

Thank you for your time and efforts.

Best regards,

Authors of Submission 3847

---

### Author Response · Authors · 2025-12-01

Dear Area Chair,

We thank the reviewers for their constructive feedback and for recognizing the importance of our work on enabling elastic inference for MoE models.

Throughout the rebuttal process, we have diligently addressed all reviewer concerns through detailed clarifications, additional experimental results, and revisions to our manuscript.

We would like to highlight our discussion with Reviewer tUR1. The reviewer's initial low rating was based on a key misunderstanding of our deployment scenario. Our clarification resolved this, leading the reviewer to raise the rating to 4 (Nov 25). To further strengthen the paper, the reviewer suggested adding a mechanistic explanation for why M-MoE maintains its elasticity. We have addressed this valuable suggestion by adding a new discussion section in the revised manuscript.

Thank you for your time and careful consideration.

Best regards,

Authors of Submission 3847

---

### Note · Authors · 2026-01-26

I have read and agree with the venue's withdrawal policy on behalf of myself and my co-authors.

---

### Meta-Review · Area_Chair_rNaY · 2026-01-05

**Summary:**

This paper presents a simple mechanism to allow for elastic number of activated experts during inference, given a single pre-trained checkpoint. The basic idea is to randomize (via different heuristics) the number of activate experts $k$ during training, so that the training and inference would not see a significant mismatch when inference is elastic. Experimental results have been presented on CPT and pre-training scenarios.

The reviewers

Overall, the proposed method is quite simple: it is based on intuitive heuristics and lacks theoretical insights. Some questions remain regarding the empirical evidence (some have not been brought up by the reviewers)

* As mentioned by Reviewer kZjX, the CPT setup is questionable. It is not clear what's the impact on model quality when pre-training a MoE with one k, and CPT with another k
  * More importantly, I share similar concern as Reviewer tUR1 & g7C7 that $k=1$ is not a common setup. It may even be wrong given your formulation. If you use softmax gate as indicated by Eq 2, $k=1$ will cause no effective gradient flowing into the router, making the whole "mixture" meaningless.
* The accuracy results in Tables 1, 2, 3 seem strange. According to the description in Section 4, all models have 96 total experts per layer, meaning that for the baselines, active parameters for $k=6$ is only 6x of $k=1$. However, the average accuracy for $k=6$ is very close to that of $k=1$ (e.g., 57.03 vs. 54.23). This may indicate some issue in the training pipeline.

**Reviewer Concerns:**

# Reviewer WfL3

**Training throughput**: authors addressed this point by additional measurement results

**Incremental compared with literature**: authors provided conceptual clarification. However, I agree that the overall method is incremental.


# Reviewer kZjX

**Comparison with dynamically-activated MoEs**: authors provided clarification on different application scenarios (elasticity in inference only)

**Perplexity & loss**: authors added those datapoints in the revision

**Alternative training strategy**: not implemented due to resources

**CPT vs. pre-training**: I agree that this is a valid concern. See comments in "Summary"


# Reviewer g7C7

**Experiments under different settings (e.g., inf. k)**: authors addressed the concerns by additional results

**Pre-training with k=1**: authors provided explanation. However, $k=1$ may be a problematic setting. See "Summary" above.

**Training throughput and cost**: authors addressed this point by additional experimental measurements

**Load-balance**: provided in Appendix F


# Reviewer tUR1

Two rounds of discussion have taken place during rebuttal. In the end, there are some remaining concerns. I echo with the reviewer that it is meaningful to rule out the scenario of "useless & harmless" experts. I consider this as a valid concern especially given the small accuracy difference even when increasing the budget of active parameters by 6x (see "Summary" above).

The reviewer also brought up valid concerns on the training configuration (e.g., k=1, see above), which remains unaddressed.

**Reviewer Scores:**

Some reviewers may moderately increase their scores after seeing the rebuttal. However, there still remain some critical points not fully addressed.

---

### Decision · Program_Chairs · 2026-01-26

Reject